# Infrared Thermal Imaging and Artificial Neural Networks to Screen for Wrist Fractures in Pediatrics

**Olamilekan Shobayo** [1] , **Reza Saatchi** [1,*] **and Shammi Ramlakhan** [2]

1 Department of Engineering and Mathematics, Sheffield Hallam University, City Campus, Howard Street, Sheffield S1 1WB, UK
2 Emergency Department, Sheffield Children's NHS Foundation Trust, Sheffield, Clarkson Street, Broomhall, Sheffield S10 2TH, UK
* Correspondence: r.saatchi@shu.ac.uk

**Abstract:** Paediatric wrist fractures are commonly seen injuries at emergency departments. Around 50% of the X-rays taken to identify these injuries indicate no fracture. The aim of this study was to develop a model using infrared thermal imaging (IRTI) data and multilayer perceptron (MLP) neural networks as a screening tool to assist clinicians in deciding which patients require X-ray imaging to diagnose a fracture. Forty participants with wrist injury (19 with a fracture, 21 without, X-ray confirmed), mean age 10.50 years, were included. IRTI of both wrists was performed with the contralateral as reference. The injured wrist region of interest (ROI) was segmented and represented by the means of cells of $10 \times 10$ pixels. The fifty largest means were selected, the mean temperature of the contralateral ROI was subtracted, and they were expressed by their standard deviation, kurtosis, and interquartile range for MLP processing. Training and test files were created, consisting of randomly split 2/3 and 1/3 of the participants, respectively. To avoid bias of participant inclusion in the two files, the experiments were repeated 100 times, and the MLP outputs were averaged. The model's sensitivity and specificity were 84.2% and 71.4%, respectively. Further work involves a larger sample size, adults, and other bone fractures.

**Keywords:** bone fracture screening; paediatrics; artificial intelligence medical diagnosis; medical infrared imaging; infrared image processing; infrared imaging feature extraction

## 1. Introduction

The wrist is made up of eight irregularly shaped carpal bones interposed between the forearm bones (radius and ulna) and five metacarpal bones. The carpal bones are arranged in two rows and consist of scaphoid, lunate, triquetrum, pisiform, hamate, capitate, trapezoid, and trapezium [1]. Wrist fractures are amongst the most common fractures in children [2] with distal radius accounting for up to 25% of fractures [3–5]. X-ray radiography is the main imaging modality for diagnosing wrist fractures [6]. A fracture is caused by the bone's structural failure due to effects such as tension, rotation, and shear [7]. Multiple factors can influence the location and type of wrist fracture. For example, distal radial fractures are typically caused by falling on an outstretched hand and can be broadly categorized as buckle, greenstick, complete, and physeal fractures [8]. After distal radius, scaphoid fractures are the second most common wrist fracture type and can be difficult to detect and treat [9], as X-ray radiographs are sometimes unremarkable. There are some less common wrist fractures, e.g., ulnar or radial styloid.

Following a fracture, direct bony union or primary fracture healing occurs with stability between the fracture surfaces while secondary fracture healing occurs with relative stability, e.g., fractures treated by a plaster cast or an external fixation [10]. Secondary fracture healing is the more common type and occurs in four overlapping phases, i.e., hematoma, inflammation, repair, and remodelling [10].

A significant proportion of X-ray radiographs for wrist fracture diagnosis fail to demonstrate a fracture. A study of 1223 children with wrist trauma reported that 51% had a wrist fracture and the rest had normal radiographs [11]. A tool which allows for rapid, non-irradiating and easy to use screening to assist clinicians in deciding which patients require X-ray radiography could be beneficial in reducing the number of unnecessary X-rays, time spent in the emergency department (ED) and associated costs.

Infrared (IR) thermal imaging (IRTI) is a well-established technology for condition monitoring in the industry; however, its applications for medical diagnosis and monitoring are currently evolving [12]. IR is part of the electromagnetic spectrum, radiated from objects with a temperature above absolute zero, i.e., $-273.15\ °C$ (0 Kelvin). Its harmless nature and ability to indicate temperatures very accurately in a non-contact manner have made it a technology of growing interest for medical diagnosis and monitoring [13]. There is evidence of an increase in temperature of the surrounding tissues at the site of the bone fracture due to increases in metabolism and blood flow [14]. This temperature increase can be quantified, analysed, and interpreted through IR thermal image processing.

To the best of our knowledge, this the first study developing IRTI and artificial neural networks to screen for wrist fractures in paediatrics. Its contributions include:

- A new method of IRTI feature extraction to suitably represent the fracture site.
- Demonstration of a statistically significant temperature difference between wrist fracture and wrist sprain (no fracture).
- Development of a multilayer perceptron (MLP) neural network model to discriminate between wrist fracture and wrist sprain.
- Effective utilization of available patient data through random selection of participants for inclusion in the training and test files for MLP processing and averaging the results over 100 trials to obtain sensitivity and specificity.

In the following sections, an overview of earlier studies related to applications of IRTI to detect or monitor bone fracture is provided, a brief description of the statistics used to determine the efficacy of the method is included, the methodology is explained, and results are discussed.

## 2. IR Thermal Imaging for Bone Fracture Detection and Monitoring

In a study involving 25 patients (mean age $65.9 \pm 10.4$ years), the mean temperature difference between a healthy and fractured distal forearm was compared [14]. They reported a mean temperature difference of $1.20 \pm 0.48\ °C$ one week after fracture, $1.42 \pm 0.54\ °C$ three weeks after fracture, $1.04 \pm 0.53°$ five weeks after fracture, $0.50 \pm 0.30\ °C$ eleven weeks after fracture and $0.22 \pm 0.25\ °C$ twenty-three weeks after fracture. Based on these findings, a fracture causes a temperature increase around the injury site that persists for several days after the injury.

IRTI was used to compare temperature difference between forearm fractures and contralateral (uninjured) side in 19 children aged 4 to 14 years [15]. The mean temperature difference across the children after 1 day was $0.13\ °C$, after 1 week was $1.17\ °C$, after 2 weeks was $0.83\ °C$, after 3 weeks was $0.23\ °C$ and after 1 month was $0.14\ °C$. The highest temperature difference occurred after 1 week. These temperature increases could be associated with the periosteal reaction at the site of fracture and the temperature decreases as the local periosteal reaction decreases and fibrous callus forms [15].

An evaluation of IRTI to detect vertebral fracture in 11 children, aged 5–18 years, with osteogenesis imperfecta (a condition causing the bones to be more fragile) was carried out [16]. The modalities to confirm vertebral fracture were dual-energy X-ray absorptiometry (DXA) and X-ray radiography. The skin temperatures above the vertebrae were compared to an area of the skin adjacent to them, acting as temperature reference. Fractured thoracic vertebrae had a significantly higher temperature compared with the reference skin temperature, while for healthy thoracic vertebrae, the temperature differences were not statistically significant.

In an earlier study, a statistical analysis of IRTI of injured wrists (fracture and sprain) was carried out to establish whether their temperatures were significantly different to the contralateral (uninjured) wrists [17]. Forty children, mean age 10.5 years (standard deviation 2.63 years), 19 with wrist fracture and 21 with wrist sprain, were recruited. Injury type (fracture or non-fracture) was determined by X-ray radiography. The mean temperature of the fractured wrists was 1.52% higher than the uninjured wrist. Although the temperature of sprained wrists was 0.97% higher than their uninjured control, the increase was not statistically significant. Similarly, IRTI analysis of 113 children aged 1 to 14 years, diagnosed with traumatic injury, indicated that the method has potential in ruling out fractures [18].

## 3. Materials and Methods

In this section, the details of evaluation statistics, recruitment, IR thermal image recordings, image processing, feature extraction, MLP pattern recognition and statistical analysis are provided.

### 3.1. Evaluation Statistics

This section briefly outlines the statistical measures used to analyse effectiveness of the IRTI method in differentiating between wrist fracture and sprain. These measures are illustrated in Figure 1, and further related information can be found in [19,20]. In the analysis, X-ray radiography was used as the gold standard, as this is what is used in clinical practice.

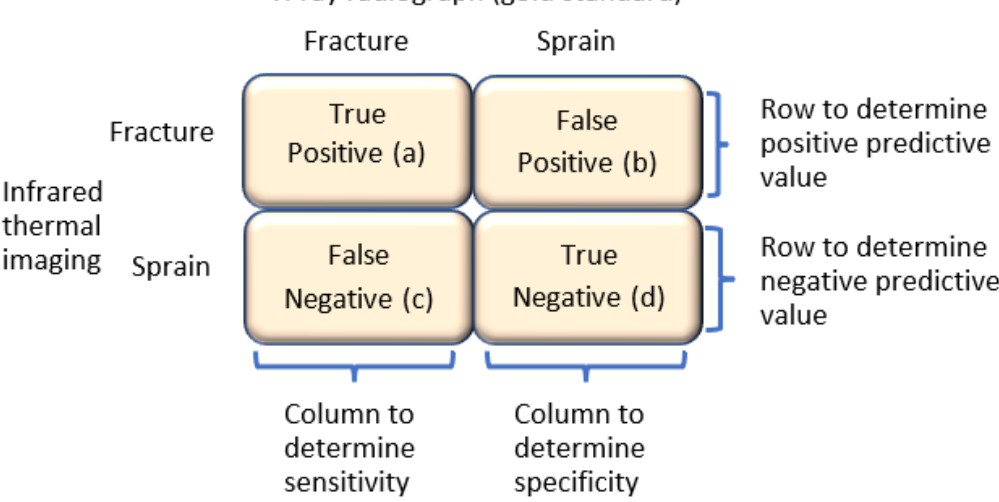

**Figure 1.** Statistical measures used to analyse effectiveness of MLP and IRTI to differentiate wrist fracture and wrist sprain.

The measures were:

- True positives, TP, (*a*): number of participants with wrist fracture (confirmed by x-ray) correctly identified as fracture by IRTI.
- False positives, FP, (*b*): number of participants with wrist sprain (not-fracture, confirmed by x-ray) misidentified as fracture by IRTI.
- False negatives, FN, (*c*): number of participants with wrist fracture misidentified as sprain by IRTI.
- True negatives, TN, (*d*): number of participants with wrist sprain correctly identified as sprain by IRTI.
- Sensitivity: the percentage of true positives (fractures) correctly identified by IRTI, i.e.,

$$Sensitivity = \left(\frac{a}{a+c}\right) \times 100 \qquad (1)$$

Specificity: the percentage of the true negatives (sprains) correctly identified by IRTI, i.e.,

$$Specificity = \left( \frac{d}{b+d} \right) \times 100 \tag{2}$$

- Positive predictive value: IRTI-identified percentage of participants with positive result (identified as fracture) who have fracture, i.e.,

$$Positive\ predictive\ value = \left( \frac{a}{a+b} \right) \times 100 \tag{3}$$

- Negative predictive value: IRTI-identified percentage of participants with a negative result (identified as sprain) who do not have fracture, i.e.,

$$Negative\ predictive\ value = \left( \frac{d}{c+d} \right) \times 100 \tag{4}$$

- Accuracy: IRTI-identified proportion of true results, either true positive or true negative, in a population. It measures the degree of veracity of IRTI as the fracture screening scheme.

$$Accuracy = \left( \frac{TP + TN}{TP + TN + FP + FN} \right) \times 100 \tag{5}$$

### 3.2. Recruitment

In total, 40 participants, 19 with wrist fracture and 21 with wrist sprain, were included in the study. This was a planned further analysis of 40 participants whose data were presented in an earlier study [17]. Participants' recruitment details are briefly provided in this section, but further information is included in [17]. The study had ethical approval from a National Health Service Research Ethics Committee (Sheffield, UK, identification number: 253,940). Participants were provided with the study's information sheet and provided signed consent. There were 24 males and 16 females, mean age 10.50 years (standard deviation 2.63 years), and 30 participants had medication (mainly paracetamol, Ibuprofen, to reduce the injury pain). Their mean body temperature was 36.3 °C (standard deviation 0.43 °C). The average time between the fracture occurrence and hospital attendance was 23.70 h (standard deviation 35.51) and for sprain 37.15 h (standard deviation 58.71 h). The following patients were excluded:

- Non-native English speakers (the study did not utilize interpreters).
- Patients sustaining multiple injuries (including injury to both wrists).
- Patients triaged above category D due to severe pain or deformity.
- Patients who declined consent.

Some patients had used ice on the injury site (to reduce pain and swelling), or their sleeves covered their wrists when attending the hospital. They have not been included in this study, as these activities would alter IR data due to non-injury mechanisms; however, in future studies, we will explore their inclusion and analysis to establish whether they still could be correctly identified.

### 3.3. Recording

A FLIR T630sc handheld IR thermal camera [21] was used for the recordings. Its specifications are: noise equivalent temperature difference (a measure indicating the minimum temperature difference resolvable by the IR camera) less than 30 mK, image resolution 640 × 480 pixels, spectral range 7.5 to 13 μm, dynamic range 14 bits, and operating temperature −40 to 650 °C (−40 °F to 1202 °F). The camera was connected to a laptop computer to facilitate initial storage of the recorded data (the recorded data were transferred to a more secure storage afterwards). Image capture rate was set to 30 frames per second (i.e., maximum rate for 640 × 480 pixels resolution) and emissivity to 0.97. The selected emissivity is suitable for recording from human skin [22]. A 10-second video of both wrists

was taken with the camera positioned above the wrists at about a meter. A video was recorded instead of a single image to allow for averaging across the resulting 300 images to reduce thermal noise. The IR thermal image recordings took place in a tertiary paediatric emergency department. Draught and external heat sources in the recording room were minimized as much as practically possible. The room temperature at the time of the data recording was measured using an accurate digital thermometer. The recording room temperature was within the recommended 18 to 25 °C range [23]. Participants remained in the recording room for 10 min prior to the recording to allow for acclimatization to room temperature. For the recordings, the participant sat comfortably on a chair with hands pronated on a thermoneutral mat placed on a table. The mat insulated their hands from possible temperature effect of the table. Matlab© version 2022 [24] (signal processing, image processing and statistical toolboxes) was used to process the recorded videos, carry out the statistical analysis and perform MLP pattern recognition.

### 3.4. Image Processing and Feature Extraction

In this section, the procedures to select the injury region of interest (ROI), track it, and represent it by distinguishing features are explained.

### 3.4.1. Selection of Region of Interest and Tracking

The procedure for selecting the ROI conformed to an earlier study that provides a more detailed explanation [17]. Matlab© package was used to display the first image of the recorded video. Using its cropping function, a region that included the carpal bones and a section of the distal radius and ulna was selected. This region is indicated by dotted lines in Figures 2 and 3 for fractured and sprained left wrists, respectively. In Figure 2, the left wrist region appears brighter (indicating increased temperature) as compared to the contralateral (uninjured) right wrist.

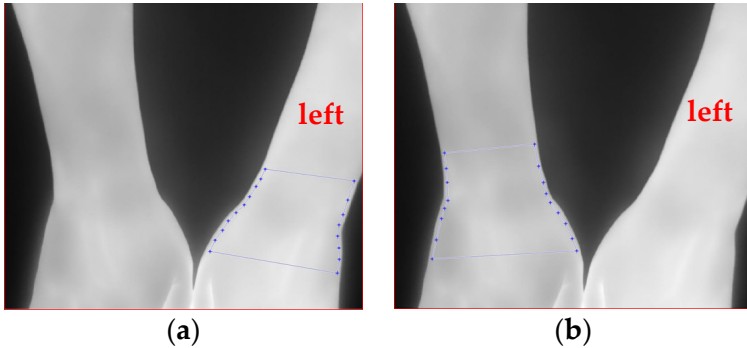

(a)                                              (b)

**Figure 2.** IR thermal images indicating the region of interest as a dotted line: (**a**) fractured left wrist; (**b**) contralateral uninjured right wrist.

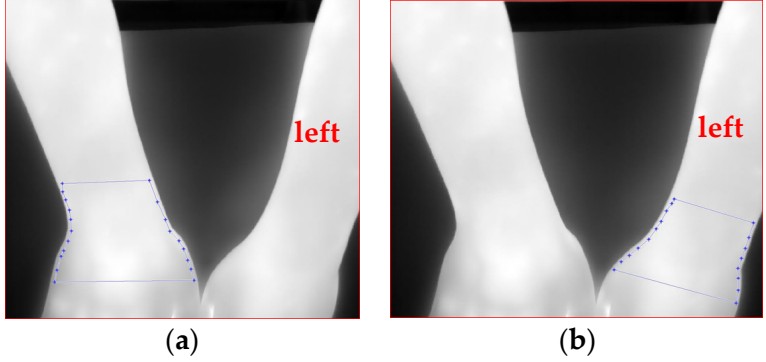

(a)                                              (b)

**Figure 3.** IR thermal images indicating the region of interest as a dotted line: (**a**) sprained left wrist; (**b**) contralateral uninjured right wrist.

Any hand movements during the recording misaligned the ROI selected from the first image and the corresponding region in the following 299 images. A template matching tracking method was applied to realign the ROI across all images. Template matching measures the similarity between two images based on their normalized cross correlation [25] and has been reported to be more accurate than methods such as the sum of absolute difference and sum of squared difference [26].

Once the ROIs from all images in a recording were extracted and aligned, they were averaged to produce a single ROI image. This process was repeated for both the injured and uninjured wrists. Figure 4 shows a typical averaged ROI. Averaging of the images enhanced them by reducing the effect of thermal noise.

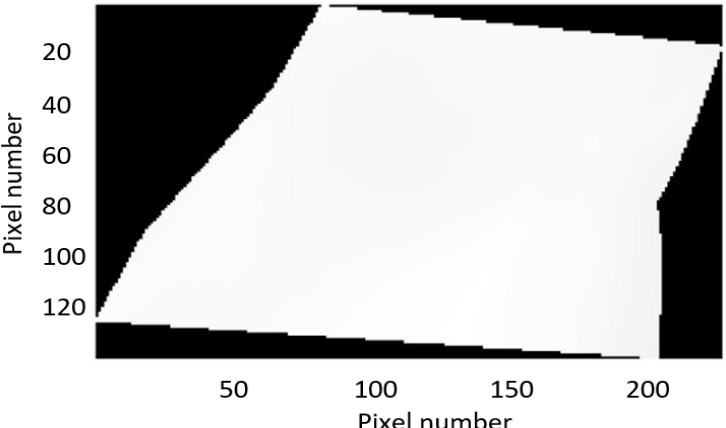

**Figure 4.** A typical averaged region of interest.

### 3.4.2. ROI Feature Extraction

The averaged ROI needed expressing by representative features for analysis by the MLP neural network. For the contralateral (uninjured wrist acting as reference temperature), initially, the background section of the ROI (shown black in Figure 4) was excluded through thresholding. The background region had zero values, and the threshold level was set accordingly. The remaining pixel values representing the wrist temperature were averaged to determine an overall reference temperature.

For the injured wrist (fractured and sprained), initially, the averaged ROI was converted to a grid, consisting of cells of $10 \times 10$ pixels. The dimension of the cells was a compromise between coarseness, representing a larger region by each cell and finer resolution allowing for a greater spatial characterization. Each cell was then represented by its mean temperature. Figure 5 shows an ROI in grid form.

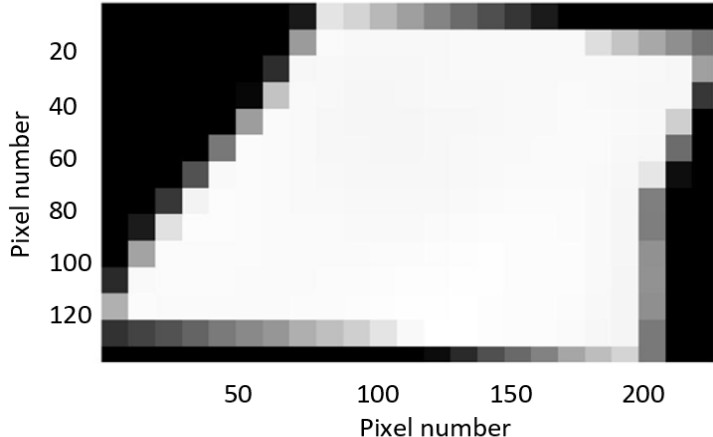

**Figure 5.** The region of interest of Figure 4 converted to cells of $10 \times 10$ pixels, and each cell represented by its mean temperature value.

The cells with greater relative temperatures were more likely to be associated with the injury location within the segmented wrist ROI. The mean temperature values for the cells were sorted in descending order of magnitude. Experiments were carried out by selecting different numbers of largest temperature values and observing the MLP discrimination results. This led to the selection of the 50 largest values (the total number of cells within an ROI varied depending on the participant as the wrist sizes varied). Selection of more than 50 values meant broadening the area within the ROI for input to the MLP, resulting in inclusion of possible areas not associated with the injury site. A smaller number of values could have caused the injury site being inadequately represented. The mean temperature representing the contralateral wrist was subtracted from the selected 50 values. This was carried out to deal with the skin temperature variations across the participants. The statistics of the resulting 50 values were obtained by considering their maximum, minimum, mean, standard deviation, median, mode, skewness, kurtosis, and interquartile range (IQR). The above operations were repeated for the 40 participants.

The justification for converting the injured wrist ROI into a grid form was that an injured wrist ROI had multiple areas with distinctly higher temperatures from its remaining parts. Initially, attempts were made to locate the injury site through clustering of the ROI; however, the approach was not effective, as multiple clusters formed, and it was unclear which cluster represented the injury site. The grid structure and selection and 50 highest temperature values ensured that the injury site was included in the analysis while the relatively cooler areas (not associated with the injury) were excluded. The approach of averaging the pixel values across the whole ROI for the injured wrist was also considered and proven not as effective, as it diminished the effect of temperature increase at the injury site by including pixels not in the vicinity of the injury site.

### 3.5. Discrimination Using Multilayer Perceptron Neural Network

MLP is a well-established artificial neural network capable of pattern recognition [27–29]. It does not assume its input data to be from a particular type of distribution or to be linearly separable. It consists of interconnected processing elements (also known as neurons) arranged as input, hidden and output layers. An MLP (shown in Figure 6) with a single hidden layer was used to differentiate between fracture and sprain based on the features extracted from the IRTI.

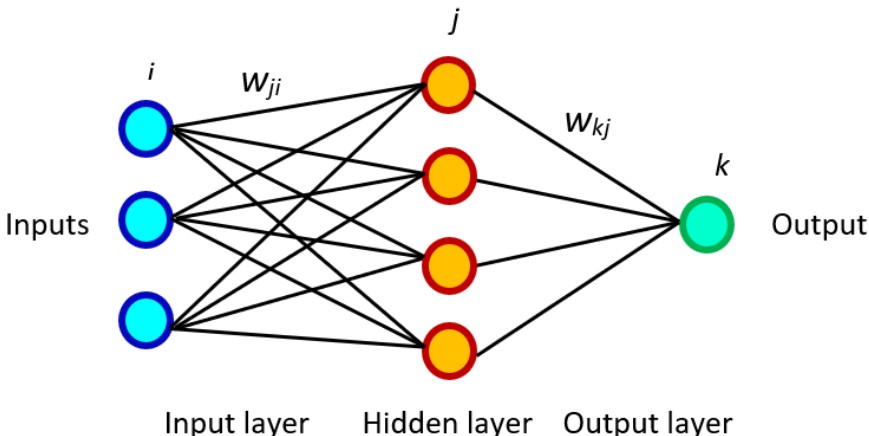

**Figure 6.** The multilayer perceptron used to differentiate wrist fracture and wrist sprain.

The MLP connections have associated weights that are initially set to random values during training. The weight for the connection to the processing element $j$ from the processing element $i$ is expressed as $w_{ji}$ in Figure 6. Similarly, the weight for the connection to the processing element $k$ from the processing element $j$ is expressed as $w_{kj}$. An MLP learns through multiple iterations using its training file, and then, its performance is evaluated using a test file that contains examples not included in the training file. The operation of an MLP is explained in articles such as [30], but it is very briefly explained here. For each

processing element, its associated inputs are multiplied with the corresponding connection weights, and the products are summed. The output of a processing element is obtained by feeding the summed value into a transfer (activation) function. The are several possible transfer functions; however, sigmoid [31] is commonly used for this purpose. During MLP training, the output of the processing element at the output layer is compared with the provided desired output (fracture = 1, sprain = 0), and the difference is used as the error. The backpropagated learning algorithm [32] is then used to reduce the error iteratively by adjusting the weights associated with the connections. A brief introduction to error backpropagation is included in Appendix A.

The MLP used in this study had three inputs (thus three processing elements at its input layer) representing standard deviation from the mean temperature, kurtosis, and interquartile range (IQR) of the 50 selected temperature values (adjusted by subtracting the mean temperature of contralateral ROI). Kurtosis quantifies how heavily the tails of a distribution differ from the tails of a normal distribution. The IQR describes the middle 50% of measures, ordered from lowest to highest. It is the difference between the upper quartile and the lower quartile in an ordered set of measures. The results section provides justification for selecting these three statistical measures. The MLP had four processing elements for its hidden layer. There is not a specific formula to calculate the optimum number of processing elements for this layer, and thus, the decision was based on experimenting with varied number of processing elements. An excessive number of hidden layer processing elements causes poor generalization (i.e., inadequate performance in correctly identifying participants not included in the training file). Not sufficient processing elements for the hidden layer cause inadequate training. The processing element at the MLP output layer had a range of 0 to 1 (0 representing sprain and 1 as fracture). The receiver operating characteristic (ROC) [33] was used to determine the boundary threshold in this range to differentiate between sprain and fracture. This is further discussed in the results. The MLP training parameters were:

- Error backpropagation learning function to update the weights: gradient descent with momentum. This learning function is commonly used with MLP. The function incorporated two parameters: learning rate and momentum. Learning rate controls its adaptation (learning or training) speed. The momentum term helps the function to move out of local minima to a global minimum when determining error [34]. For both parameters, values between 0.01 and 1 were explored, and 0.05 was selected, as it provided the best differentiation.
- Training termination: Training stopped the duration of each trial when the error became insignificant (0.01) or when the number of iterations reached 20,000. The second criteria ensured training to be terminated when the error could not reach its specified target value.
- Transfer (activation function): the sigmoid transfer function was used for all processing elements. It provides an output between 0 and 1 and is commonly used for MLP [31].

The range (minimum to maximum values) for each measure used as input to the MLP varied. To ensure that the differences in the range of measures did not adversely influence their relative contributions to the MLP output, each measure was individually mapped so that its minimum value corresponded to zero, and its maximum value corresponded to one. Other values were scaled accordingly. The formula used for this purpose was

$$x_s = \frac{x - Minimum\ value}{Maximum\ value - Minimum\ value} \tag{6}$$

where $x_s$ and $x$ represent scaled values and original values used as inputs to the MLP. This scaling also ensured that the inputs to MLP conformed to the range of the sigmoid transfer function (i.e., 0 to 1).

Given the available sample size, a strategy had to be devised to maximize the scope of the MLP training and accuracy of its performance during evaluation. A limited sample size can often occur in medical machine learning scenarios [35,36], and thus, strategies to

make effective use of the available data were reported [37]. In a study, multiple runs of a single instance of a neural network were trained, each run having variations in some training settings, and collective statistics were generated [38]. The issue of small sample size to diagnose glaucoma was dealt with by applying a few-shot learning approach [39]. Oversampling was applied to balance the number of available examples when applying machine learning for chronic kidney disease risk predication [40]. Another approach is data augmentation, where the amount of data are artificially increased by fine adjustments (e.g., flipping, rotation, cropping, zooming, noise addition, colour transformations) of the available data or creating synthetic data [41]. The approach has been applied for deep learning of images [42–45]. Deep learning has a feature learning capability; however, for the MLP used in this study, a process of feature preparation was undertaken. MLP has a lower data processing capability than deep learning, but it is much less computationally intensive. Therefore, as compared to deep learning, MLP can be applied more quickly and can utilize hardware platforms with lower processing capabilities. The extent an augmentation process may help improving the effectiveness of the MLP in this study (given the complexities of the IR images) is not certain and could be an area of future exploration. This study approached the limited sample size by adapting two investigations.

Investigation A: This involved creation of a training file consisting of 27 randomly selected participants (i.e., 2/3 of the 40 participants) and placing the remaining 13 participants (1/3 of the participants) in a test (evaluation) file. The MLP was trained on the participants in the training file and evaluated on the participants in the test file. The training and test files were again regenerated in a similar manner, and the MLP training and evaluation were repeated. This operation was performed 100 times, and overall, MLP outputs for the test files were averaged. This procedure ensured that there was no bias in selecting specific participants for each of the two files, as the repetitions (trials) resulted in all participants having an opportunity to be included in both the training and tests files.

Investigation B: This was like investigation A, except the number of participants included in the training file was increased to 35, and the number of participants included in the test file was reduced to 5. The purpose of this investigation was to explore the effect of increasing the number of participants in the training file on the discrimination accuracy of the MLP. In some circumstances, an increase in the training file size may have a positive effect on MLP training, as more participants are examined during each trial. However, its downside is that the MLP evaluation would be on a reduced number of participants.

## 4. Results

In this section, the features representing the wrist ROI are analysed, and the MLP performance in discriminating between wrist fracture and sprain is presented.

### 4.1. Feature Analysis

Average differences between the injured and contralateral uninjured wrists for the statistical measures are provided in Table 1. These are also shown as boxplots in Figure 7.

**Table 1.** Average differences between the injured and contralateral uninjured (reference) wrists for the statistical measures, the percentage difference and number of participants differentiated by the statistical measures (fracture (f), sprain (s)).

| | Maximum (°C) | Minimum (°C) | Mean (°C) | Std Dev. (°C) | Median (°C) | Mode (°C) | Skewness | Kurtosis | IQR (°C) |
|---|---|---|---|---|---|---|---|---|---|
| Fracture | 1.396 | 0.696 | 0.962 | 0.187 | 0.938 | 0.696 | 0.408 | 2.478 | 0.272 |
| Sprain | 1.048 | 0.530 | 0.711 | 0.136 | 0.690 | 0.530 | 0.595 | 2.804 | 0.202 |
| %Difference | 24.942 | 23.873 | 26.076 | 27.322 | 26.439 | 23.873 | −45.931 | 13.157 | 25.752 |
| Number of participants differentiated | 13 (f > s) | 10 (f > s) | 12 (f > s) | 14 (f > s) | 12 (f > s) | 8 (f > s) | 13 (f < s) | 16 (f < s) | 14 (f > s) |

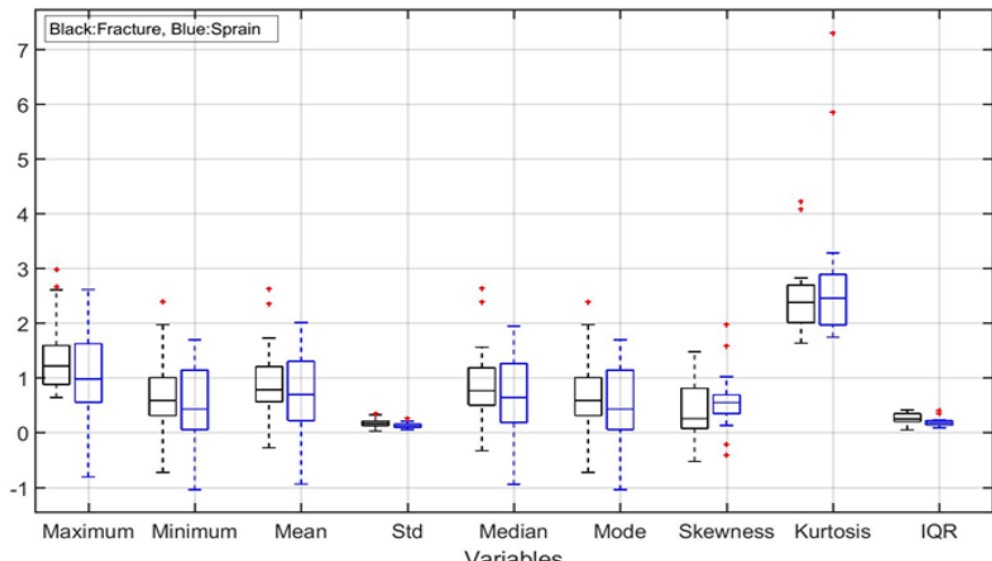

**Figure 7.** Boxplots of average differences between the injured and uninjured contralateral wrists for the statistical measures. Note: the label for vertical axis is not shown, as the variables do not have the same unit. The unit for minimum, maximum, mean, standard deviation (std) and interquartile range (IQR) is °C. Skewness and kurtosis have no units.

To select the statistical measures that provided greater differentiation between fracture and sprain, the following operations were performed. The mean values of the statistical measures for the sprained wrists were obtained. These values were compared with the values of the associated statistical measure for the fractured wrists. The following observations were made:

- 13 (68.4%) participants with fracture had maximum temperatures greater than sprain participants maximum temperature.
- 10 (52.6%) participants with fracture had minimum temperatures greater than sprain participants minimum temperature.
- 12 (63.2%) participants with fracture had mean temperatures greater than sprain participants mean temperature.
- 14 (73.7%) participants with fracture had standard deviations (from the mean) greater than sprain participants standard deviation.
- 12 (63.2%) participants with fracture had median temperatures greater than sprain participants median temperature.
- 8 (42.1%) participants with fracture had mode temperatures greater than the sprain participants mode temperature.
- 13 (68.4%) participants with fracture had skewness values lower than sprain participants skewness.
- 16 (82.2%) participants with fracture had kurtosis values lower than sprain participants kurtosis.
- 14 (73.7%) participants with fracture had IQR values greater than sprain participants IQR.

The above analysis indicated that the measures' effectiveness to differentiate between the two types of injuries in descending order were:

- Kurtosis;
- Standard deviation from the mean and IQR;
- Skewness and maximum temperature;
- Mean and median;
- Minimum;
- Mode.

For the measures that were indicative of temperature magnitude (i.e., maximum, minimum, mean, standard deviation, median, mode, IQR), most participants with a fracture

had values greater than respective measures for sprain. However, for the measures that were indicative of distribution (i.e., kurtosis and skewness), the opposite was the case.

The above analysis led to the selection of kurtosis, standard deviation from the mean, and IQR for input to the MLP. Inclusion of other measures did not improve the MLP differentiation results, and thus, they were omitted.

### 4.2. Multilayer Perceptron Discrimination Results for Investigation A

A statistical summary of investigation A, indicating the averaged (over 100 trials) MLP outputs for the participants in the test file, is provided in Table 2. The averaged MLP outputs for fracture and sprain were 0.589 and 0.349, respectively.

**Table 2.** Investigation A—averaged multilayer perceptron outputs for participants in the test file (averaged over 100 trials). The values have no units.

| Injury Types | Average | Standard Deviation |
| --- | --- | --- |
| Fracture | 0.589 | 0.264 |
| Sprain | 0.349 | 0.247 |

To explore the MLP differentiation effectiveness, the percentage difference (PD) and percentage absolute difference (PAD) were obtained. The formulae used for these were

$$PD = \frac{F - S}{F} \times 100 \tag{7}$$

$$PAD = \left| \frac{F - S}{0.5(F - S)} \right| \times 100 \tag{8}$$

where

$F$ = MLP output averaged over 100 trials for participants with fracture included in the test file;

$S$ = MLP output averaged over 100 trials for participants with sprain included in the test file.

The *PD* and *PAD* values were 40.75% and 45.02%, respectively.

To determine the classification boundary between fracture and sprain, the receiver operating characteristic curve (ROC) was obtained for the averaged MLP outputs (over 100 trials) for the participants in the test file. The resulting plot is shown in Figure 8.

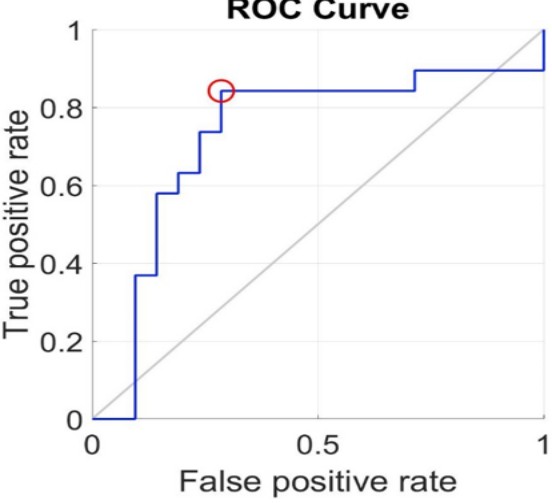

**Figure 8.** The receiver operating characteristic curve (ROC) for investigation A.

The area under ROC was 0.727, and the ROC optimum point for discrimination for the two types of injuries corresponded to a false positive rate = 0.289 and true positive

rate = 0.842 resulting in the threshold value differentiating fracture and sprain to be 0.380. Using this threshold, the averaged MLP output for participants in the test file was analysed, resulting in:

- Number of true positives = 16;
- Number of true negatives = 15;
- Number of false negatives = 3;
- Number of false positives = 6.

These measures resulted in discrimination sensitivity of 0.842 (84.2%) and specificity of 0.714 (71.4%). The positive and negative predictive values were 0.723 and 0.833, respectively. The overall discrimination accuracy was 77.5%.

The plot of averaged MLP outputs for participants in the test file are shown in Figure 9. The threshold level is shown as the horizontal dashed line at 0.380.

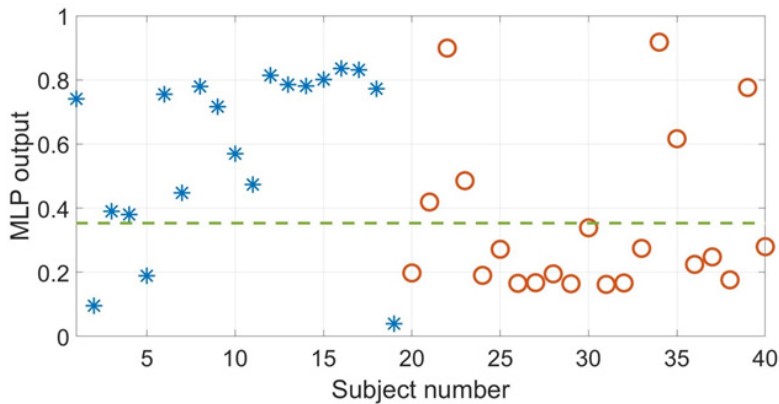

**Figure 9.** Plot of averaged multilayer perceptron outputs (over 100 trials) for participants in the test file in investigation A. The stars and circles represent participants with fracture and sprain, respectively. The dashed line is the threshold.

Figure 9 shows three participants with fracture appearing under the threshold line with sprain participants and six participants with sprain appearing above the threshold line with fracture participants.

*4.3. Multilayer Perceptron Discrimination Results for Investigation B*

A statistical summary of investigation B indicating the averaged (over 100 trials) MLP outputs for the participants in the test file is provided in Table 3. The average MLP outputs for fracture and sprain were 0.617 and 0.345, respectively.

**Table 3.** Averaged multilayer perceptron outputs (over 100 trials) for participants in the test file, investigation B. The values have no units.

| Injury Types | Average | Standard Deviation |
|---|---|---|
| Fracture | 0.617 | 0.280 |
| Sprain | 0.345 | 0.252 |

The PD and PAD values for this investigation were 44.084% and 56.549%, respectively.

To determine the classification boundary between fracture and sprain, the receiver operating characteristic curve (ROC) was obtained for the averaged MLP values (over 100 trials) for the participants in the test file. The resulting plot is shown in Figure 10.

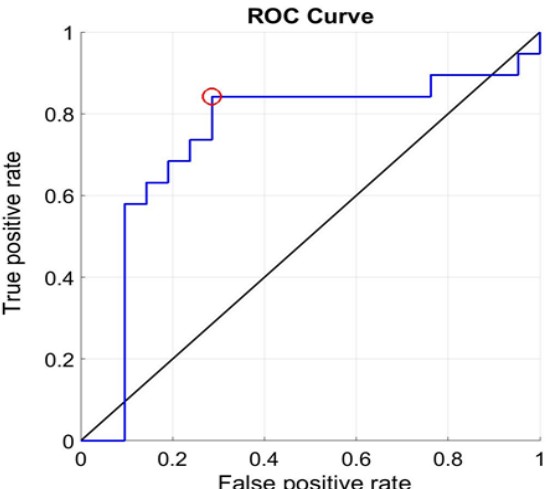

**Figure 10.** The receiver operating characteristic curve (ROC) for investigation B.

The area under ROC was 0.742, and the ROC optimum point was associated with a false positive rate = 0.286 and true positive rate = 0.842. These corresponded to a differentiation threshold value of 0.353. Using this threshold, the MLP outputs for participants in the test file were analysed, providing:

- Number of true positives = 16;
- Number of true negatives = 15;
- Number of false negatives = 3;
- Number of false positives = 6.

These values resulted in differentiation sensitivity of 0.842 (84.2%) and specificity of 0.714 (71.4%). The positive and negative predictive values were 0.727 and 0.833, respectively. The overall discrimination accuracy was 77.5%.

The averaged MLP outputs for participants in the test file are shown in Figure 11. The threshold level is shown as the horizontal dashed line at 0.353.

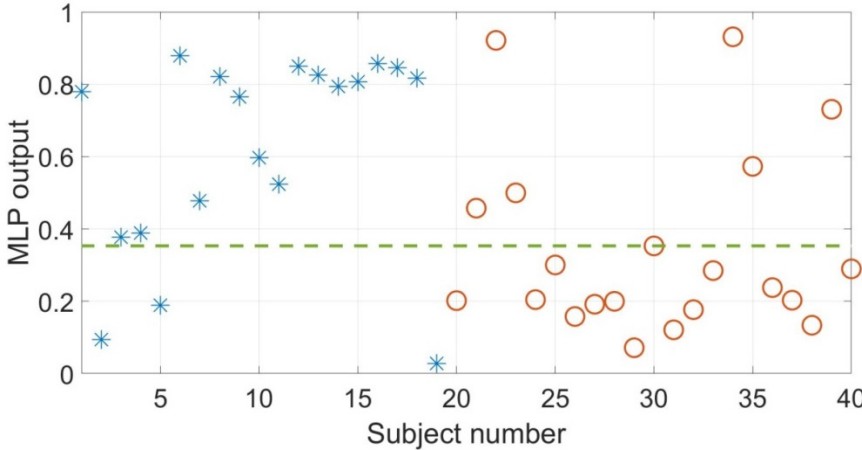

**Figure 11.** Plot of averaged multilayer perceptron outputs for participants in the test file for investigation B. The stars and circles represent participants with fracture and sprain, respectively. The dashed line is the threshold.

## 5. Discussion

A fracture causes a temperature increase at the site of injury that is significantly higher than that produced by a sprain [17]. This study investigated the effectiveness of MLP neural networks in using this effect to screen for wrist fractures. The ability of an MLP in differentiation tasks is affected by the manner of data pre-processing and feature

extraction. The grid representation of the ROI for injured wrists allowed pixels spatially close to each other with similar temperature values to be expressed by their mean values. This then allowed the 50 largest mean temperature values from the injured wrist ROI to be selected for characterization by statistical measures. To deal with skin temperature variability across the participants, the mean temperature of the contralateral (uninjured) ROI was subtracted from the selected values. Each cell within the grid was represented by $10 \times 10$ pixels, i.e., total 100 pixels. The cell size was determined by experimenting with different dimensions and observing the MLP differentiation outcomes. In future studies, a more in-depth analysis could be carried out in determining the cell size. The statistical measures used to analyse the 50 selected values, quantified the magnitude of the temperature (i.e., maximum, minimum, mean, mode and median), temperature deviation from their mean value, interquartile range and distribution (i.e., skewness and kurtosis). Standard deviation from the mean, kurtosis, and interquartile range (IQR) proved more effective for differentiating between the two types of injuries and thus were used as input to the MLP. The selection of these three measures should be considered in the context of the limited number of participants included in this study, and thus, with a larger number of participants, the effectiveness of these three measures needs further evaluation.

The manner of randomly selecting participants for MLP training and test files and averaging the MLP outputs over 100 trials was to mitigate the bias that can occur when selecting patients for each file. This bias may alter the results when the sample size is not large. MLP can differentiate between nonlinearly separable data and is not sensitive to the type of data distribution. These can make MLP a more robust classifier than statistical methods such as linear discriminant analysis.

The discrimination sensitivity (84.2%) and specificity (71.4%) values indicated that the method was more effective in correctly identifying the participants with fracture (16 participants from 19) than correctly identifying participants with sprain (15 participants from 21). Comparison of the results from investigations A and B indicated that changes in the number of participants in the training and test files had not affected the sensitivity and specificity.

The severity of injury types (fracture and sprain) varied across the participants. It is possible that a severe sprain may cause a greater temperature increase at the site of the injury than a small fracture. Furthermore, this study did not consider the time between the occurrence of the injury and the IRTI recordings. This may have negatively affected the MLP performance and thus requires exploration in a follow-up study. False negatives (fracture classified as sprain) have a more serious impact on treating patients than false positives (sprain classified as fracture). There were three false negatives and six false positive cases. The method being developed is for screening patients for requesting X-ray radiography. Therefore, false positive cases should be picked up by the follow-up X-ray radiographs. The tool is not to replace X-ray radiography as the gold standard fracture diagnostic but to assist clinicians at the triage stage to identify patients more effectively. The study focused on paediatric wrist fractures, and follow-up work should also include other fractures and expand to include adults.

The main limitation of the study was the small number of participants that would have adversely affected the sensitivity and specificity values. The study excluded participants who had applied ice to the injury site prior to hospital attendance and those who had their wrists covered with sleeves. Follow-up studies should examine whether those patients could be accurately screened using the method outlined.

## 6. Conclusions

The study explored the effectiveness of multilayer perceptron (MLP) and IR thermal imaging (IRTI) to screen for paediatric wrist fractures. A grid structure-based method of representing injured wrists for feature extraction was devised and characterized by statistical measures for input to the MLP. Two investigations were carried out involving different numbers of participants in the training and test files. The sensitivity and specificity obtained by both methods were consistent (84.2% and 71.4%, respectively). The overall

accuracy in both methods was 77.5%. The main limitation of the study was the sample size (i.e., 19 with wrist fracture and 21 with wrist sprain). A larger database is likely to improve the differentiation of specificity and specificity. The study indicated that application of MLP to suitably selected IRTI features could have potential for screening for wrist fractures in paediatrics. The method could be explored further for other fractures and in adults.

**Author Contributions:** Conceptualization, O.S., R.S. and S.R.; methodology, O.S., R.S. and S.R.; validation, O.S., R.S. and S.R.; investigation, O.S., R.S. and S.R.; data curation, O.S., R.S. and S.R.; writing—original draft preparation, O.S., R.S. and S.R.; writing—review and editing, O.S., R.S. and S.R. All authors have read and agreed to the published version of the manuscript.

**Funding:** This research received no external funding.

**Institutional Review Board Statement:** The study was conducted in accordance with the Declaration of Helsinki, and approved by the National Health Service Research Ethics Committee (United Kingdom, identification number: 253,940, approval date: 7 March 2019. Informed consent was obtained from all subjects involved in the study.

**Data Availability Statement:** Due to ethical restrictions, the study's data will not be shared.

**Acknowledgments:** The authors are very grateful to Charlotte Reed for assistance in the recording of the data used in the study. They are also grateful to all the children who took part in the study and for the cooperation of their carers.

**Conflicts of Interest:** The authors declare no conflict of interest.

## Appendix A  Error Backpropagation Algorithm

Error backpropagation is a well-known learning algorithm for multilayer perceptron (MLP) artificial neural networks, and its theory is explained in several articles, e.g., [32,46]. The knowledge of its derivation is not essential in understanding the work reported in the article, but for completeness, it is very briefly introduced. Referring to Figure 6:

- The subscripts *i*, *j* and *k* represent the input, hidden and output layers of the MLP, respectively.
- The weights from the hidden layer to the output layer: $w_{kj}$;
- The weights from the input layer to the hidden layer: $w_{ji}$;
- The input to a processing element: *net*;
- The output of a processing element (i.e., the transfer function output): *y*;
- The target (desired) value provided during training: *t*;
- Number of input examples used during training: *k*;
- The convergence control parameter (learning rate): $\epsilon$;
- Proportionality: $\propto$.

The study utilized gradient descent for the backpropagation learning algorithm. The sum-squared error (*E*) is

$$E = 0.5 \sum_k (t_k - y_k)^2 \tag{A1}$$

The algorithm reduces the overall error *E* during training by iteratively updating the network's weights. The amount of change $\Delta W$ is proportional to the rate of change of *E* with respect to the weights (negative sign is needed to reduce *E*), i.e.,

$$\Delta W \propto -\frac{\partial E}{\partial W} \tag{A2}$$

The amount of change for the weights from the hidden layer to the output layer is determined by

$$\Delta w_{kj} \propto -\frac{\partial E}{\partial \Delta w_{kj}} = -\epsilon \frac{\partial E}{\partial y_k} \frac{\partial y_k}{\partial net_k} \frac{\partial net_k}{\partial w_{kj}} \tag{A3}$$

However,

$$\frac{\partial E}{\partial y_k} = \frac{\partial\left(0.5(t_k - y_k)^2\right)}{\partial y_k} = -(t_k - y_k) \tag{A4}$$

For the sigmoid transfer function used in this study,

$$\frac{\partial y_k}{\partial net_k} = \frac{\partial\left(1 + e^{-net_k}\right)^{-1}}{\partial net_k} = \frac{e^{-net_k}}{\left(1 + e^{-net_k}\right)^2} = y_k(1 - y_k) \tag{A5}$$

$$\frac{\partial net_k}{\partial w_{kj}} = \frac{\partial\left(w_{kj}\, y_j\right)}{\partial w_{kj}} = y_j \tag{A6}$$

$$\Delta w_{kj} = \epsilon(t_k - y_k)y_k(1 - y_k)y_j = \epsilon\delta_k y_j \tag{A7}$$

where

$$\delta_k = (t_k - y_k)y_k(1 - y_k) \tag{A8}$$

The amount of change for the weights from the input layer to the hidden layer is determined by

$$\Delta w_{ji} \propto -\left[\frac{\partial E}{\partial y_k} \frac{\partial y_k}{\partial net_k} \frac{\partial net_k}{\partial y_j}\right] \frac{\partial y_j}{\partial net_j} \frac{\partial net_j}{\partial w_{ji}} \tag{A9}$$

$$\Delta w_{ji} = \epsilon\left[\sum_k \delta_k w_{kj}\right] y_j(1 - y_j)y_i = \epsilon\delta_j y_i \tag{A10}$$

where

$$\delta_j = \left[\sum_k \delta_k w_{kj}\right] y_j(1 - y_j) \tag{A11}$$

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
