# Peer review of "Infrared Thermal Imaging and Artificial Neural Networks to Screen for Wrist Fractures in Pediatrics"

_technologies, doi:10.3390/technologies10060119_

Round 1

Reviewer 1 Report

The paper deals with important task. The authors considered a pediatric wrist fractures identification task.

The paper has great practical value.

Suggestions:

1.       It would be good to add point-by-point the main contributions at the end of the Introduction section

2.       The authord should add a strong Related works section. As tha main problem of this paper is size of dataset, please analyzed existing works that working with small datasets. The authors can stress attention on “Input-doubling methods” and  “Additive input-doubling methods”   among others

3.       It is unclear why the authors didn’t use data augmentation section.

4.       The authors should provide a link to open access repository with the dataset used for modeling

5.       The authors should add all optimal parameters for all investigated methods

Author Response

Dear Honorary Editors, Respected reviewers

Thank you very much for so kindly reviewing our article and making very valuable constructive comments. We have very carefully considered your comments and have done our best to amend and improve the paper according to your suggestions. The changes made are summarised in the attached table  and are highlighted blue on the article.   

We are very grateful for the help and support provided and hope our revisions meet your expectations.

Best wishes

Professor Reza Saatchi

Reviewer 2 Report

The article entitled “Infrared thermal imaging and artificial neural networks to screen for wrist fractures in pediatrics” is well-written and, from my point of view, would be of interest for the readers of Technologies. Despite of this and before its publication, I would recommend the following changes to be performed:

Introduction: is it known by the authors any similar study to the one here presented that would be cited? If so, indicate and if not let readers also know.

The concepts that are defined in the section entitled Evaluation Statistics are well-known therefore, think about if such section would be removed or summarized. Please also note that Evaluation Statistics should be included in Materials and Methods.

The content of 4.4. Discrimination Using Multilayer Perceptron Neural Network ia naïve and lacks of mathematical background. Please improve it. Please also provide details about the real implementation employed in your research. Which software? Which library?

Line 386 please, explain the interest of the measurement of Kurtosis in this context.

Equation (8): please use the mathematical symbol of absolute value instead of the word absolute.

Author Response

(The authors gave the same response as above.)

Round 2

Reviewer 1 Report

Paper can be accepted in current form

Reviewer 2 Report

After the changes performed by the authors I consider that the paper is ready for its publication.  Congratulations.